# The Lattice Representation Hypothesis of Large Language Models

**Bo Xiong**
Stanford University, United States
`xiongbo@stanford.edu`

## Abstract

We propose the *Lattice Representation Hypothesis* of large language models: a symbolic backbone that grounds conceptual hierarchies and logical operations in embedding geometry. Our framework unifies the *Linear Representation Hypothesis* with *Formal Concept Analysis (FCA)*, showing that linear attribute directions with separating thresholds induce a concept lattice via half-space intersections. This geometry enables symbolic reasoning through geometric meet (*intersection*) and join (*union*) operations, and admits a canonical form when attribute directions are linearly independent. Experiments on WordNet provide empirical evidence that LLM embeddings encode concept lattices and their logical structure, revealing a principled bridge between continuous geometry and symbolic abstraction.

## 1 Introduction

Large language models (LLMs) (Achiam et al., 2023; Grattafiori et al., 2024; Mesnard et al., 2024) are surprisingly effective in capturing conceptual knowledge (Petroni et al., 2019; Wu et al., 2023; Lin & Ng, 2022; Xiong & Staab, 2025) and performing logical reasoning, capabilities traditionally associated with symbolic AI. Yet, it remains fundamentally unclear how exactly such conceptual knowledge, including concepts, hierarchies, and their logical semantics, is encoded within the continuous geometry of LLM representation spaces. Unlocking this hidden geometry is crucial not only for interpreting how LLMS representing symbolic knowledge, but also for reliably controlling and steering their inference behavior (Han et al., 2024), a fundamental step for advancing AI safety.

To understand concept representations in LLMs, a promising direction is the *Linear Representation Hypothesis* (Mikolov et al., 2013b; Park et al., 2025; 2024a; Gurnee & Tegmark, 2024), which posits that semantic features and concepts are encoded as linear directions or subspaces in a model's embedding space. This idea, rooted in early work on word embeddings (Pennington et al., 2014a), has since been extended to modern LLMs, where such directions can be interpreted as embedding difference, logistic probing, or steering vectors in different contexts (Gurnee & Tegmark, 2024; Nanda et al., 2023; Zhao et al., 2025). Park et al. (Park et al., 2024a) unifies them through *causal inner product* that respects the semantic structure of concepts in the sense that causally separable concepts are represented by orthogonal vectors. However, these works mainly focus on exploring the existence of the linearity of (binary) concepts, but offer limited insights for interpreting compositional or set-theoretic semantics such as concept inclusion, concept intersection, and union, which lies at the heart of symbolic abstraction.

Recently, Park et al. (Park et al., 2025) extended the Linear Representation Hypothesis to formalize *categorical concepts* as geometric regions like *polytopes* in the representation space, and show that semantic hierarchy corresponds to orthogonality. However, they model concepts purely in terms of their extensions, that is, as sets of tokens or objects that fall under the category, such as $Y(\texttt{animal}) = \{\texttt{predator}, \texttt{bird}, \texttt{dog}, ...\}$. While this extensional view is useful for evaluating membership, it overlooks their intensional nature, i.e., the attributes and relations that ground categories in logic and philosophy, making it difficult to interpret how concepts are related to each others through set-theoretic semantics like concept subsumption, intersection, or union.

We draw inspiration from Formal Concept Analysis (FCA) (Ganter et al., 2005), a principled and philosophy-inspired framework that defines concepts through both their instances and their attributes. In FCA, each concept is represented as a pair: an extent (the set of objects) and an intent (the set of

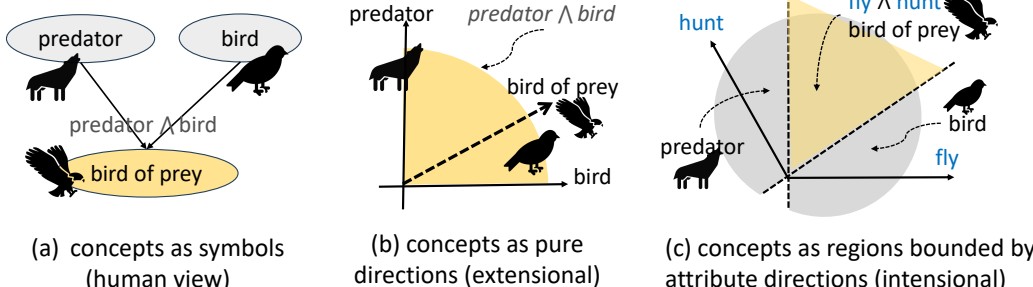

(a) concepts as symbols (human view)

(b) concepts as pure directions (extensional)

(c) concepts as regions bounded by attribute directions (intensional)

Figure 1: **How LLMs encode conceptual structure.** (a) Humans represent concepts as symbols and compose them using logical operators. (b) Under the standard extensional view, LLMs encode concepts as directions, where subsumption is interpreted through the relative orientation of vectors. (c) Under our intensional view, a concept is represented as the intersection of half-spaces defined by its attributes, and compositional semantics emerges through region intersection and union.

shared attributes). For example, the concept bird may be defined by attributes such as can fly, has feathers, and lays eggs, while eagle (a bird of prey) refines this category by denoting a subset of birds with additional features such as can hunt. As Figure 1 shows, unlike extensional view defining concepts as instances, such dual intent-extent view defines concepts as convex regions bounded by their attributes defining them. For example, the concept *bird* may be defined by attributes like *can fly*, *has feathers*, and *lays eggs*, while *eagle* (a bird of prey) refines this category by denoting a subset of birds with additional features such as *can hunt*. This dual formulation naturally induces a concept lattice, because every pair of concepts can be ordered by inclusion of their extents (or, dually, their intents), and any two concepts have a well-defined intersection (their common subconcept) and union (their common superconcept), corresponding to the meet and join operations of the lattice.

In this work, we propose the *Lattice Representation Hypothesis* that unifies two perspectives on concept representation: the *Linear Representation Hypothesis*, which views semantics as directions in embedding geometry, and *Formal Concept Analysis*, which formalizes concepts through the incidence relation between objects and attributes. **Our key insight is that these views coincide, revealing a hidden lattice geometry in LLMs:** attribute directions correspond to FCA intents, object embeddings to extents, and symbolic abstractions such as subsumption, intersection, and union emerge naturally from the induced closure structure. Building on this connection, we formalize a half-space model of concepts and a projection-based notion of concept inclusion that together recover a lattice geometry from LLM representations. Our contributions are threefold: (i) a theoretical framework linking Linear Representation Hypothesis to FCA via half-space intersections, (ii) a soft inclusion measure and concept algebra (meet/join) defined directly on embeddings, and (iii) empirical evidence on WordNet sub-hierarchies showing that LLM embeddings encode concept lattices, enabling coherent generalization (join) and refinement (meet). These results demonstrate that LLMs implicitly organize conceptual knowledge into a lattice geometry, providing a symbolic backbone for interpretability.

## 2 PRELIMINARIES

### 2.1 LINEAR REPRESENTATION HYPOTHESIS IN LLMs

We consider the autoregressive family of LLMs that predict the next tokens given its context.

**Definition 1** (Large Language Model). *An LLM defines a probability distribution over the next token $y$ given a context $x$ via the softmax function $\Pr(y \mid x) \propto \exp\left(\lambda(x)^\top \gamma(y)\right)$, where $\lambda : \mathcal{X} \to \Lambda \simeq \mathbb{R}^d$ maps the input context $x$ to a context embedding vector $\lambda(x)$, and $\gamma : \mathcal{V} \to \Gamma \simeq \mathbb{R}^d$ maps each vocabulary token $y$ to its unembedding vector $\gamma(y)$.*

This definition involves a *context embedding space* $\Lambda$ and a *token unembedding space* $\Gamma$, which together define the geometry of the softmax distribution. These two spaces can be unified via the *causal inner product* (Park et al., 2024a). Specifically, there exists an invertible matrix $A \in \mathbb{R}^{d \times d}$ and a constant vector $\bar{\gamma}_0 \in \mathbb{R}^d$ such that defining $g(y) := A\left(\gamma(y) - \bar{\gamma}_0\right)$ and $\ell(x) := A^{-\top}\lambda(x)$, reparameterizes token and context embeddings into a shared semantic space, where their interaction

is captured by the Euclidean inner product $\ell(x)^\top g(y)$. This transformation preserves the model's output distribution, as the softmax $\Pr(y \mid x)$ remains invariant under any choice of $A$ and $\bar{\gamma}_0$.

In this unified space, Linear Representation Hypothesis states that certain semantic attributes correspond to specific directions. There is an explicit definition for binary attributes.

**Definition 2** (Linear representation of a binary attribute/concept (Park et al., 2024a)). *A vector $\bar{\ell}_m \in \mathbb{R}^d$ is said to linearly represent a binary attribute $m \in \{0, 1\}$ if, for all context embeddings $\ell \in \mathbb{R}^d$, all scalars $\alpha > 0$, and all attributes $z \neq m$ that are causally separable from $m$, the following conditions hold:*

- *Attribute activation:* $\Pr(m = 1 \mid \ell + \alpha\bar{\ell}_m) > \Pr(m = 1 \mid \ell)$;

- *Causal selectivity:* $\Pr(z \mid \ell + \alpha\bar{\ell}_m) = \Pr(z \mid \ell)$.

In other words, moving in the direction $\bar{\ell}_m$ increases the likelihood of the attribute $m$ without affecting any causally unrelated attributes. The direction merely encodes the semantic direction of the attribute, not its strength, as any positive scalar multiple $\alpha\bar{\ell}_m$ has the same qualitative effect.

## 2.2 FORMAL CONCEPT ANALYSIS (FCA)

FCA (Ganter et al., 2003) is a mathematical framework for modeling concepts as structured relationships between objects and their attributes. Unlike extensional views that define concepts simply as sets of objects (as in (Cowsik et al., 2024)), FCA treats a concept as an intensional abstraction: a set of objects characterized by a common set of attributes. This duality between objects and attributes allows FCA to capture both the semantic content of a concept and its compositional structure. FCA begins with a binary relation between objects and attributes, formalized as a *formal context*:

**Definition 3** (Formal context). *A formal context is a triple $(G, M, I)$, where $G$ is a finite set of objects, $M$ is a finite set of attributes, and $I \subseteq G \times M$ is a binary relation (called the incidence relation) such that $(g, m) \in I$ indicates that object $g \in G$ possesses attribute $m \in M$.*

From this, FCA defines concepts as maximal sets of objects and attributes that are mutually consistent:

**Definition 4** (Formal concept). *Given a formal context $(G, M, I)$, consider a pair $(A, B)$ with $A \subseteq G$ and $B \subseteq M$. Define the Galois connections as $A' := \{m \in M \mid \forall g \in A, (g, m) \in I\}$, $B' := \{g \in G \mid \forall m \in B, (g, m) \in I\}$. The pair $(A, B)$ is called a* formal concept *if and only if $A' = B$ and $B' = A$, where $A$ is the* extent *and $B$ is the* intent.

The set of formal concepts is partially ordered by inclusion of extents (i.e., $A_1 \subseteq A_2$) or equivalently by reverse inclusion of intents (i.e., $B_2 \subseteq B_1$).

**Definition 5** (Concept lattice). *Let $(A_1, B_1)$ and $(A_2, B_2)$ be formal concepts. Then $(A_1, B_1) \leq_C (A_2, B_2)$ if and only if $A_1 \subseteq A_2$ (equivalently, $B_2 \subseteq B_1$), where $\leq_C$ denotes the partial order relationship. Under the partial order $\leq_C$, the set of all formal concepts forms a complete lattice: every subset concepts have a greatest lower bound (meet) and a least upper bound (join).*

## 3 THE LATTICE REPRESENTATION GEOMETRY IN LLMS

We now establish a connection between the Linear Representation Hypothesis and Formal Concept Analysis (FCA), showing that the linear geometry of LLM embeddings gives rise to a concept lattice. The key idea is that each binary attribute can be modeled as a direction in the unified space, with membership approximated by a thresholded inner product. This naturally induces a binary object–attribute relation, from which a formal context and concept lattice can be constructed. We refer to this construction as the *lattice geometry* of LLMs. Fig. 2 illustrates such correspondence.

### 3.1 FROM LINEAR TO LATTICE GEOMETRY

**Geometric interpretation of attribute membership.** Under the Linear Representation Hypothesis, semantic attributes are encoded as directions while objects (tokens or contexts) are encoded as vectors. Let $\bar{\ell}_m \in \mathbb{R}^d$ denote the direction for attribute $m$, and $\mathbf{v}_g \in \mathbb{R}^d$ the embedding of object $g$. The extent to which $g$ possesses $m$ can be estimated by the projection $\mathbf{v}_g \cdot \bar{\ell}_m$. In the idealized case, there

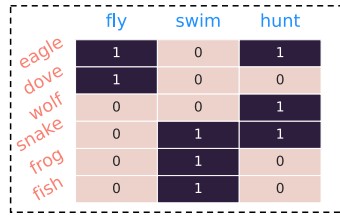 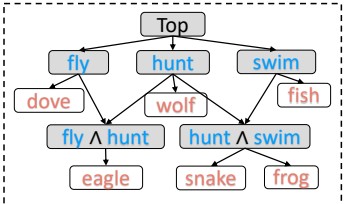 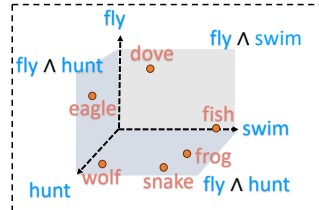

(a) Object-attribute relation
(formal context)

(b) Discrete lattice
constructed via FCA

(c) Lattice geometry linearly
encoded in LLMs

Figure 2: **How FCA connects to the linear lattice geometry of LLMs.** (a) A formal context describing which **objects** satisfy which **attributes**. (b) The discrete concept lattice constructed exactly from this formal context. (c) The corresponding lattice geometry encoded in LLM embeddings, where each attribute is represented as a linear direction and each object as a point, and concept composition (meet and join) emerges as intersection or union of half-spaces.

exists a threshold $\tau_m$ such that $m(g) = 1 \iff \mathbf{v}_g \cdot \bar{\ell}_m \geq \tau_m$. However, perfect separation rarely holds in practice, so we define a *soft incidence relation*:

$$P_\alpha(m(g) = 1) := \sigma\big(\alpha \cdot (\mathbf{v}_g \cdot \bar{\ell}_m - \tau_m)\big), \tag{1}$$

where $\alpha > 0$ is a sharpness parameter controlling the smoothness of the logistic incidence function. A larger $\alpha$ makes the boundary steeper and smaller $\alpha$ makes the boundary smoother. As $\alpha \to \infty$, this recovers the hard-thresholded case. This assigns each object–attribute pair a fuzzy degree of membership, enabling smoother definitions of concepts.

**Theorem 1** (Existence of Lattice Geometry). *Let $G$ be a finite set of objects and $M$ a finite set of attributes. Let $V = \{\mathbf{v}_g \in \mathbb{R}^d \mid g \in G\}$ be object embeddings and $\mathcal{D} = \{\bar{\ell}_m \in \mathbb{R}^d \mid m \in M\}$ attribute directions. Suppose for each $m \in M$ there exists a threshold $\tau_m \in \mathbb{R}$ such that membership is modeled by the soft incidence function above. For any confidence level $\delta \in (0, 1)$, define the binary incidence relation*

$$I_\delta := \{(\mathbf{v}_g, \bar{\ell}_m) \mid P_\alpha(m(g) = 1) \geq \delta\}.$$

*Then the induced concept set*

$$\mathcal{F}_\delta = \left\{ (X, Y) \,\middle|\, X = \{\mathbf{v} \in V \mid \forall \bar{\ell} \in Y, (\mathbf{v}, \bar{\ell}) \in I_\delta\}, \, Y = \{\bar{\ell} \in \mathcal{D} \mid \forall \mathbf{v} \in X, (\mathbf{v}, \bar{\ell}) \in I_\delta\} \right\}$$

*satisfies: (i) closure under the Galois connection, and (ii) forms a complete lattice under extent inclusion (equivalently, reverse intent inclusion). Proof is detailed in Appendix B.*

Thus, when attributes are encoded as thresholded linear projections, a symbolic concept lattice can be recovered from embedding geometry, capturing semantic abstraction through graded boundaries.

**Canonical representation under soft incidence.** Theorem 1 allows arbitrary thresholds. We now show that under mild conditions, these thresholds can be absorbed into a global shift of embeddings, yielding a canonical origin-passing form:

**Proposition 1** (Canonical representation). *Let each attribute $m_i \in M$ be defined by direction $\mathbf{d}_i$ and threshold $\tau_i$. Let $D \in \mathbb{R}^{k \times d}$ be the matrix with rows $\mathbf{d}_i^\top$, and $\boldsymbol{\tau} = (\tau_1, \ldots, \tau_k)$. If there exists $\mathbf{c} \in \mathbb{R}^d$ such that $D\mathbf{c} = \boldsymbol{\tau}$, then*

$$\sigma(\alpha(\mathbf{v}_g \cdot \mathbf{d}_i - \tau_i)) = \sigma(\alpha((\mathbf{v}_g - \mathbf{c}) \cdot \mathbf{d}_i)) \quad \forall g, i. \tag{2}$$

That is, the probabilities remain invariant under the transformation $\mathbf{v}_g \mapsto \mathbf{v}_g - \mathbf{c}$, reducing the model to a canonical half-space form while preserving the induced lattice. Proof is detailed in Appendix C.

## 3.2 HALF-SPACE MODEL AND CONCEPT ALGEBRA

Under the canonical representation, where all attribute thresholds have been absorbed via a global shift, each attribute defines an origin-passing half-space in the embedding space. A concept composed of multiple attributes can be interpreted geometrically as the intersection of those half-spaces, i.e., the region where all attribute constraints are simultaneously satisfied.

**Definition 6** (Concept as half-space). *Let $M$ be a set of attributes, each represented by a direction $\mathbf{d}_m \in \mathbb{R}^d$. A concept defined by a subset $Y \subseteq M$ corresponds to the set of object embeddings that satisfy all associated directional constraints:*

$$\mathcal{R}(Y) := \left\{ \mathbf{v} \in \mathbb{R}^d \mid \mathbf{v} \cdot \mathbf{d}_m \geq 0 \text{ for all } m \in Y \right\}. \tag{3}$$

Geometrically, $\mathcal{R}(Y)$ is a convex polyhedral cone defined by intersecting origin-passing half-spaces. However, this definition assumes perfect attribute separation, which may not hold in practice due to noise, uncertainty, and overlapping concept boundaries in LLM representations. To address this, we move from a hard region view to a *soft formulation*, where membership is modeled as graded alignment rather than binary inclusion.

**Concept representation via normalized projection profiles.** Given a concept $C$ (e.g., a lexical term) and its associated context embeddings $\{\mathbf{v}_1, \ldots, \mathbf{v}_n\} \subset \mathbb{R}^d$, we define its semantic representation using the average projection profile over the attribute directions $\{\mathbf{d}_m\}_{m \in M}$. For each attribute $m \in M$, the projection value is

$$\pi_C(m) := \frac{1}{n} \sum_{i=1}^{n} \mathbf{v}_i \cdot \mathbf{d}_m. \tag{4}$$

The resulting projection vector $\boldsymbol{\pi}_C \in \mathbb{R}^{|M|}$ encodes a *soft attribute profile* of $C$, reflecting how strongly the concept aligns with each attribute. This can be seen as a continuous analogue of an FCA intent. To ensure comparability across concepts, all projection vectors are $\ell_2$-normalized.

**Concept inclusion as region containment.** With projection profiles in place, we can now define a graded notion of subsumption between two concepts. Intuitively, a concept $A$ is included in another concept $B$ if the attributes emphasized by $B$ are also strongly expressed in $A$. We capture this by a *soft inclusion score* that evaluates how well $A$'s profile satisfies the attribute activations of $B$:

$$\text{Inclusion}(A \sqsubseteq B) = \frac{\sum_{m \in M} \phi(\pi_B(m)) \cdot \sigma(\pi_A(m))}{\sum_{m \in M} \phi(\pi_B(m))}, \quad \text{where} \quad \phi(x) = \log(1 + e^x). \tag{5}$$

Here, the sigmoid $\sigma(\cdot)$ maps $A$'s projection value to a soft likelihood of satisfying attribute $m$, while the softplus $\phi(\cdot)$ weights attributes according to their salience in $B$. This formulation smoothly downweights weakly expressed or inactive attributes in $B$, while strongly positive ones dominate the inclusion score. Thus, concept inclusion is modeled not as a strict set-theoretic containment but as a continuous, geometry-driven compatibility measure between attribute profiles.

**Concept Meet and Join.** We operationalize concept algebra in the half-space model by defining meet and join directly on concept regions $\mathcal{R}(Y)$. Meet is the intersection of regions, while join is the least region subsuming both concepts, approximated by the conic hull of their defining directions.

**Definition 7** (Concept algebra: meet and join). *Let $\mathcal{R}(Y)$ denote the region associated with a concept defined by attribute set $Y \subseteq M$ (Definition 6).*

- *Meet (intersection). For two concepts $A = \mathcal{R}(Y_A)$ and $B = \mathcal{R}(Y_B)$, their meet is the region satisfying all attributes from both sets $A \wedge B := \mathcal{R}(Y_A \cup Y_B)$. Geometrically, this corresponds to intersecting the half-spaces that define $A$ and $B$.*

- *Join (union or generalization). Their join is the least upper bound in the lattice, i.e., the most specific concept region that subsumes both $A$ and $B$. In the half-space model, this corresponds to the minimal region that covers $\mathcal{R}(Y_A)$ and $\mathcal{R}(Y_B)$: $A \vee B := \mathcal{R}(Y_A) \cup \mathcal{R}(Y_B)$, which can be approximated by the conic hull spanned by the attribute directions of $A$ and $B$.*

**Soft measure of meet/join.** While Definition 7 specifies meet and join geometrically, we also require a *graded* way to evaluate how well a concept $C$ corresponds to these symbolic operations.

**Soft meet/join profiles.** Given two concepts $A, B$ with projection profiles $\pi_A, \pi_B$, we define the projection profile of their meet and join using fuzzy $t$-norm/co-norm combinations:

$$\pi_{A \wedge B}(m) = \min\{\pi_A(m), \pi_B(m)\}, \qquad \pi_{A \vee B}(m) = \max\{\pi_A(m), \pi_B(m)\}. \tag{6}$$

**Degrees of inclusion.** The degree to which $C$ is subsumed by the meet or join is

$$\deg(C \sqsubseteq A \wedge B) = \text{Inclusion}(C \sqsubseteq A \wedge B), \qquad \deg(C \sqsubseteq A \vee B) = \text{Inclusion}(C \sqsubseteq A \vee B), \quad (7)$$

where the inclusion function is as defined in Eq. (2).

**Degrees of equality.** Soft equality is defined by symmetrizing inclusion with the harmonic mean:

$$\deg(C = A \wedge B) = \frac{2\,\text{Incl}(C \sqsubseteq A \wedge B) \cdot \text{Incl}(A \wedge B \sqsubseteq C)}{\text{Incl}(C \sqsubseteq A \wedge B) + \text{Incl}(A \wedge B \sqsubseteq C)}, \tag{8}$$

$$\deg(C = A \vee B) = \frac{2\,\text{Incl}(C \sqsubseteq A \vee B) \cdot \text{Incl}(A \vee B \sqsubseteq C)}{\text{Incl}(C \sqsubseteq A \vee B) + \text{Incl}(A \vee B \sqsubseteq C)}. \tag{9}$$

## 4 EVALUATION

In this section, we empirically evaluate the extent to which the linear structure of LLM embeddings induces a lattice geometry over concepts. We focus on testing the two core assumptions underlying our framework: 1) **Existence of half-space model:** How well do the linear representations of binary attributes recover the ground-truth formal context? 2) **Existence of lattice geometry in LLMs:** Does the approximated formal context recover a valid partial order set or concept lattice?

### 4.1 DATASET AND EXPERIMENT SETUP

**Dataset construction.** We construct five object–attribute datasets derived from the WordNet hierarchy. Three datasets correspond to physical domains (**WN-Animal**, **WN-Plant**, **WN-Food**), and two correspond to abstract domains (**WN-Event**, **WN-Cognition**). Each dataset represents a distinct semantic domain. Statistics of datasets are shown in Table 4 in the Appendix. For each domain, we extract all concept terms that fall under the corresponding hierarchy using the WordNet `is_a` (hypernym) relation. We expand each concept by retrieving its synonyms and hyponyms to ensure lexical coverage and semantic granularity. Since WordNet does not provide explicit attribute annotations, we use a large language model (GPT-4o) to generate the attribute schema and populate the object–attribute matrix. Specifically, for each category, we first prompt the model to produce a concise set of salient binary attributes relevant to classification within the category (e.g., `can fly`, `has fur`, `lays eggs` for animals). We then use few-shot prompting to annotate each object with a binary attribute vector. This produces a complete binary incidence matrix, which we treat as the ground-truth *formal context* for evaluation. We use the WordNet `is_a` relation to define a symbolic subsumption hierarchy, and treat it as the target concept lattice structure. Using the annotated formal context and corresponding LLM embeddings for objects and attributes, we evaluate whether the geometry of the embedding space can reconstruct the symbolic structure implied by the context.

**Object embedding** Each object $g$ is represented by aggregating the embeddings of its lexical synonyms. For every WordNet object $g$, we retrieve all lemma names in its synset and treat them as synonymous surface forms. For each synonym string $s$, we compute its embedding $\text{Emb}(s) \in \mathbb{R}^d$ as the last hidden state of the model averaged across token positions. The final object embedding is defined as the mean over its synonym embeddings,

$$\mathbf{v}_g := \frac{1}{|\text{Syn}(g)|} \sum_{s \in \text{Syn}(g)} \text{Emb}(s), \tag{10}$$

where $\text{Syn}(g)$ denotes the set of lemma names associated with $g$. For example, the synset of *German shepherd* includes {*German shepherd*, *German shepherd dog*, *German police dog*, *alsatian*}. Averaging across these variants reduces lexical noise, stabilizes surface-form effects, and yields a more faithful representation of the underlying concept rather than a specific phrasing.

**Attribute embedding** To estimate the semantic direction associated with each attribute, we apply a linear discriminant analysis approach using object embeddings labeled as positive or negative with respect to that attribute. Given a set of positive and negative object embeddings for attribute $m$, we first compute the class means $\boldsymbol{\mu}_+$ and $\boldsymbol{\mu}_-$, and the class covariance matrices $\Sigma_+$ and $\Sigma_-$, estimated

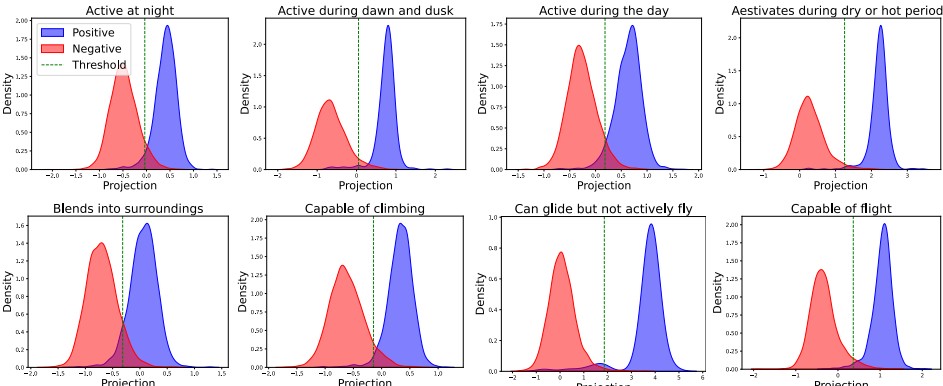

Figure 3: Distribution of projection lengths for positive and negative objects onto the directions of the first eight attributes (sorted alphabetically) in the WN-Animal dataset.

using Ledoit-Wolf shrinkage to improve robustness in high-dimensional settings. We then define the attribute direction $\bar{\ell}_m$ as the solution to a regularized Fisher separation criterion:

$$\bar{\ell}_m := (\Sigma_+ + \Sigma_- + \lambda I)^{-1} (\boldsymbol{\mu}_+ - \boldsymbol{\mu}_-), \tag{11}$$

where $\lambda > 0$ is a small regularization constant to ensure numerical stability. This procedure yields a direction vector that best separates the positive and negative object clusters in embedding space under a linear discriminant model. The resulting vector $\bar{\ell}_m$ is used as the attribute direction in all downstream projections and geometric reasoning.

**Threshold estimation** Given an attribute direction $\bar{\ell}_m$, we determine a threshold $\tau_m \in \mathbb{R}$ to separate positive and negative objects along this direction. For each object embedding $\mathbf{v}_g$, we compute its projection onto the normalized direction, and then calculate the threshold as the average of the mean projections of positive and negative object sets:

$$\tau_m := \frac{1}{2} \left( \mathbb{E}_{g \in G_+}[\text{Proj}_m(\mathbf{v}_g)] + \mathbb{E}_{g \in G_-}[\text{Proj}_m(\mathbf{v}_g)] \right), \tag{12}$$

where $G_+$ and $G_-$ denote the sets of positive and negative objects for attribute $m$, respectively. This threshold minimizes classification error under the assumption of linear separability and equal cost.

## 4.2 EXISTENCE OF HALF-SPACE MODEL

We first evaluate whether semantic attributes in LLM embedding space adhere to the half-space model, i.e., whether a single linear direction and threshold can reliably separate objects that possess a given attribute from those that do not. For each attribute $m$, we estimate a direction $\mathbf{d}_m$ and threshold $\tau_m$ using the training set (Section 4.1). Given an object embedding $\mathbf{v}_g$, we predict whether the object possesses the attribute using a hard decision rule $\hat{m}(g) = \mathbb{I}\left[\mathbf{v}_g \cdot \mathbf{d}_m \geq \tau_m\right]$, where $\mathbb{I}[\cdot]$ is the indicator function. We compare this prediction to the ground-truth incidence $m(g) \in \{0, 1\}$ from the annotated formal context. For each attribute, we compute precision, recall, and F1 score, and report averages over all attributes within each dataset.

**Results** Table 1 reports results for recovering the formal context across five WordNet domains. The *Linear* method achieves the best precision, recall, and F1 on every model and domain, consistently above 78% on physical domains (WN-Animal, WN-Plant, WN-Food) and still strong on the more abstract Event and Cognition domains (above 70%). This demonstrates that LDA-estimated attribute directions align closely with ground-truth concept–attribute structure, even in semantically diffuse domains. The *Mean* baseline performs moderately (59–68% F1), capturing coarse centroids but lacking fine-grained discriminative power. The *Random* baseline stays near chance (45–48%), confirming that attribute recovery is far from trivial and requires meaningful geometric structure. Across models, Gemma-7B and LLaMA3-8B show the strongest linear separability, with Gemma-7B reaching 83.2% F1 on Animal and Plant, and Mistral-7B following closely. Overall, these results provide clear evidence that LLM embeddings support the half-space model of concepts proposed in Section 3.2. Figure 3 further illustrates the separation between positive and negative projection distributions, showing clear margins around the estimated thresholds.

Table 1: Evaluation of recovering concept–attribute relations (formal context). **Bold** denotes the best across models, shading denotes the best within model. All metrics are percentages.

| Model | | WN-Animal | | | WN-Plant | | | WN-Food | | | WN-Event | | | WN-Cognition | | |
|---|---|---|---|---|---|---|---|---|---|---|---|---|---|---|---|---|
| | | Pre. | Rec. | F1 | Pre. | Rec. | F1 | Pre. | Rec. | F1 | Pre. | Rec. | F1 | Pre. | Rec. | F1 |
| LLaMA3.1-8B | Random | 49.7 | 49.6 | 45.3 | 50.1 | 50.1 | 47.3 | 50.0 | 50.0 | 46.4 | 50.2 | 50.2 | 48.6 | 50.2 | 50.2 | 50.1 |
| | Mean | 63.7 | 69.3 | 63.7 | 63.7 | 67.2 | 63.3 | 67.2 | 71.4 | 68.1 | 63.7 | 65.1 | 63.9 | 68.4 | 68.4 | 68.4 |
| | Linear | 81.4 | 84.0 | 82.5 | 81.6 | 82.4 | 82.4 | **79.7** | 80.6 | **80.1** | 71.4 | 71.6 | **71.5** | 75.0 | 74.9 | 75.0 |
| Gemma-7B | Random | 49.8 | 49.7 | 45.3 | 50.1 | 50.1 | 47.3 | 50.1 | 50.1 | 46.3 | 49.4 | 49.3 | 47.8 | 50.1 | 50.1 | 50.1 |
| | Mean | 53.5 | 55.2 | 50.1 | 53.5 | 54.4 | 51.3 | 53.7 | 55.1 | 51.2 | 53.3 | 53.8 | 52.2 | 56.4 | 56.4 | 56.3 |
| | Linear | **82.0** | **84.8** | **83.2** | **82.3** | **84.3** | **83.2** | 79.2 | **80.9** | 80.0 | 71.1 | **71.7** | 71.4 | **75.4** | **75.4** | **75.4** |
| Mistral-7B | Random | 49.4 | 49.2 | 45.0 | 50.3 | 50.4 | 47.5 | 49.5 | 49.5 | 45.5 | 50.5 | 50.6 | 49.0 | 49.4 | 49.4 | 49.3 |
| | Mean | 62.2 | 67.1 | 62.0 | 61.9 | 64.9 | 61.4 | 62.4 | 66.6 | 62.1 | 57.2 | 58.1 | 56.5 | 63.3 | 63.3 | 63.3 |
| | Linear | 80.6 | 83.4 | 81.8 | 80.8 | 82.8 | 81.7 | 77.6 | 78.9 | 78.2 | 69.7 | 69.8 | 69.7 | 74.2 | 74.1 | 74.1 |

Table 2: Evaluation of partial order inference from projection-based concept representations. **Bold** denotes the best across models, shading denotes the best within model. All metrics are percentages.

| Model | | WN-Animal | | | WN-Plant | | | WN-Food | | | WN-Event | | | WN-Cognition | | |
|---|---|---|---|---|---|---|---|---|---|---|---|---|---|---|---|---|
| | | Pre. | Rec. | F1 | Pre. | Rec. | F1 | Pre. | Rec. | F1 | Pre. | Rec. | F1 | Pre. | Rec. | F1 |
| LLaMA3.1-8B | Random | 52.3 | 43.2 | 47.3 | 51.1 | 49.5 | 47.6 | 25.0 | 50.0 | 33.3 | 50.2 | 50.2 | 50.2 | 49.8 | 49.9 | 49.8 |
| | Mean | 68.4 | 65.0 | 66.7 | 64.2 | 63.3 | 63.8 | 61.5 | 58.5 | 55.7 | 60.3 | 57.9 | 59.1 | 58.2 | 55.4 | 56.8 |
| | Linear | **75.9** | **78.3** | **77.1** | 71.2 | 70.0 | 70.4 | **78.1** | 75.9 | 75.4 | **69.1** | 67.5 | **68.3** | 70.4 | 68.9 | **69.6** |
| Gemma-7B | Random | 54.0 | 51.2 | 50.6 | 52.7 | 50.3 | 49.5 | 53.4 | 50.8 | 39.1 | 49.8 | 50.1 | 49.9 | 49.4 | 49.7 | 49.5 |
| | Mean | 66.2 | 61.0 | 63.4 | 62.4 | 59.8 | 60.9 | 50.7 | 50.7 | 50.6 | 56.3 | 55.0 | 55.6 | 54.1 | 52.8 | 53.4 |
| | Linear | 74.4 | 76.0 | 75.1 | **72.9** | **70.1** | **71.4** | 76.3 | 75.8 | **75.6** | 66.2 | 65.1 | 65.6 | 67.0 | 65.9 | 66.4 |
| Mistral-7B | Random | 53.3 | 45.9 | 49.3 | 50.0 | 51.1 | 48.2 | 25.0 | 50.0 | 33.3 | 49.3 | 49.2 | 49.2 | 48.7 | 49.0 | 48.8 |
| | Mean | 66.8 | 63.4 | 64.9 | 61.2 | 60.0 | 60.5 | 56.0 | 55.6 | 54.8 | 55.7 | 54.3 | 55.0 | 53.3 | 51.9 | 52.6 |
| | Linear | 71.7 | 72.6 | 72.1 | 65.7 | 60.6 | 57.1 | 68.8 | 64.3 | 62.0 | 62.9 | 60.8 | 61.8 | 62.0 | 60.3 | 61.1 |

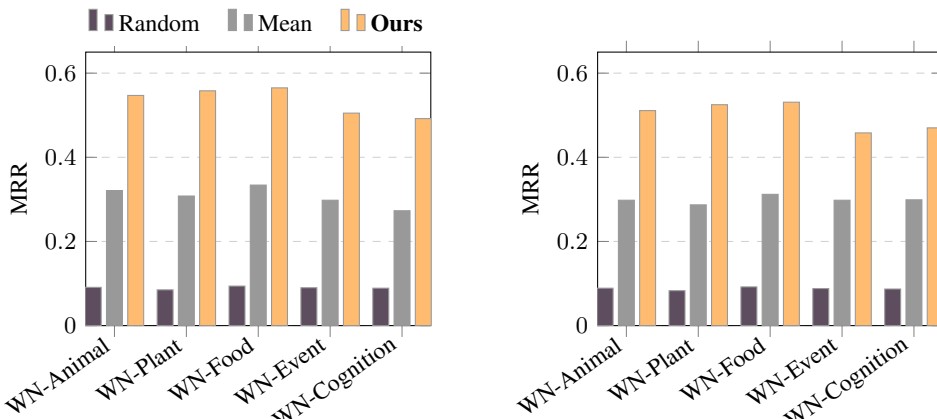

Figure 4: Quantitative evaluation (MRR) of concept algebra for Meet (*left*) and Join (*right*) operators.

## 4.3 EXISTENCE OF LATTICE GEOMETRY

To test the *existence of lattice geometry*, we use the concept inclusion score defined in Section 3.2, where projection profiles over attribute directions are compared via the soft inclusion formula. This allows us to infer subsumptions directly from embedding geometry without access to ground-truth hierarchies. As Table 2 shows, the LINEAR method consistently outperforms centroid-based (MEAN) and random baselines across all domains, achieving F1 scores up to 77.1 (LLaMA, WN-Animal) and 75.6 (Gemma, WN-Food). These gains demonstrate that discriminatively estimated attribute directions capture intensional information sufficient to recover hierarchical relations: concepts with stronger projection activation on the attributes of another concept are reliably inferred as subconcepts. The empirical alignment between projection-based subsumption and WordNet ground-truth provides strong evidence that LLM embeddings indeed admit a partial order structure, supporting our hypothesis that lattice geometry is embedded within their representation space.

Table 3: Top-10 terms related to the join and meet of selected WordNet concept pairs.

| A | B | A ∨ B | A ∧ B |
|---|---|---|---|
| dog | wolf | predator, animal, canine, meat-eater, hunter, wild, mammal, quadruped, pet | dog, hound, puppy, terrier, mutt, beagle, retriever, spaniel, shepherd, pooch |
| cat | lion | predator, feline, animal, meat-eater, beast, carnivore, hunter, whiskers, mammal, wild | cat, kitten, tiger, panther, leopard, tomcat, feline, cheetah, tabby, lynx |
| sparrow | robin | avian, songbird, fowl, feathered, finch, beak, chirp, nest, perching, small | sparrow, robin, songbird, warbler, finch, canary, thrush, chickadee, pipit, titmouse |
| horse | zebra | equid, hoofed, animal, ungulate, mammal, quadruped, stallion, beast, herbivore | horse, pony, stallion, mare, foal, filly, mustang, gelding, thoroughbred, colt |
| carrot | parsnip | root, edible, produce, food, green, vegetable, crunchy, fresh, garden, plant | carrot, parsnip, radish, beet, turnip, tuber, rootcrop, veg, sprout, crop |
| eagle | falcon | animal, raptor, bird, predator, creature, wingspan, talon, flyer, sky, sharp-eyed | eagle, falcon, hawk, osprey, kestrel, buzzard, kite, harrier, condor, bird of prey |

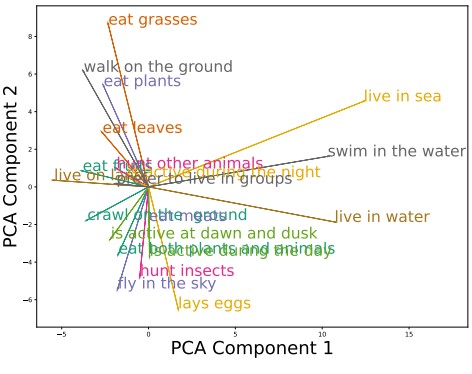
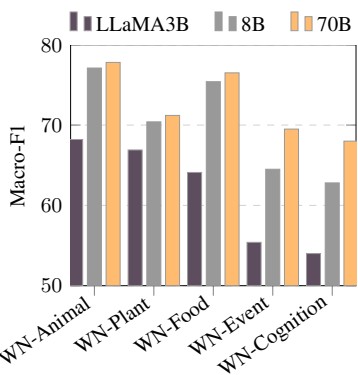

Figure 5: (a) PCA-based visualization of attribute directions in WN-Animal (top 20 most frequent attributes); (b) Performance of LLaMA-3 models of different sizes across WordNet domains.

**Quantitative evaluation of concept algebra** We quantitatively evaluate the concept algebra using the *degree of equality* metrics in Eq. 8–9. For each WordNet domain, we randomly sample 200 concept pairs (A,B) that have at least one shared descendant and one shared ancestor, ensuring well-defined symbolic meets and joins. Gold meets are the lowest shared descendants; gold joins are the least common hypernyms. We compare three approaches: a **Random** scorer, a **Mean** baseline that ranks candidates by similarity to the averaged embeddings of (A, B), and our concept-algebra operator. For each predicted meet or join, we rank all candidates and report the MRR of the gold labels. Figure 4 shows that our method consistently outperforms both baselines, with the largest improvements on physical domains and slightly lower gains on more abstract ones.

**Qualitative evaluation of concept algebra** We further qualitatively evaluate the concept algebra by randomly selecting concept pairs and inspecting their top-ranked meet and join candidates. As shown in Table 3, the *join* operation $(A \lor B)$ reliably returns higher-level abstractions subsuming both inputs (e.g., *predator* for *dog* and *wolf*, or *avian* for *sparrow* and *robin*), mirroring WordNet hypernyms. Conversely, the *meet* operation $(A \land B)$ produces refined category intersections such as *pony*/*stallion*/*foal* for *horse* and *zebra*, or *hawk*/*osprey* for *eagle* and *falcon*. These examples illustrate that meet and join behave as geometric analogues of set-theoretic conjunction and abstraction, providing qualitative evidence that LLM embeddings encode a coherent latent lattice structure suitable for compositional reasoning.

## 4.4 ADDITIONAL ANALYSIS

**Physical vs. abstract domains** Table 1, Table 2, Figure 4, and Figure 5b show a consistent tendency: physical domains (Animal, Plant, Food) achieve relatively better results than abstract domains (Event, Cognition). We conjecture that this is because physical concepts are grounded in concrete, human-perceptual attributes (shape, movement, habitat, function), while abstract concepts rely on more complicated or situational attributes that are less directly encoded in LLMs.

**Attribute correlation analysis**   We visualize the top 20 most frequent attributes in WN-Animal using PCA (Figure 5a) and report their pairwise correlations in the Appendix. The PCA plot reveals coherent semantic clusters, for example, *eat grasses* and *eat plants* appear close together, while *swim in water* and *live in the sea* form a tight group capturing aquatic behaviors. In contrast, attributes associated with unrelated ecological or behavioral properties (e.g., *lay eggs* vs. *live in water*) lie far apart in both the PCA and correlation map. These patterns indicate that the learned attribute directions naturally organize into meaningful semantic subspaces.

**Effect of model scaling**   We also analyze how lattice-geometry performance scales with model size by comparing LLaMA-3 models from 3B to 70B parameters. As shown in Figure 5b, scaling leads to consistent but modest gains on physical domains (WN-Animal, WN-Plant, WN-Food), where even smaller models already capture grounded attributes that structure these categories. In contrast, scaling yields much larger improvements on abstract domains (WN-Event, WN-Cognition), where success depends on representing non-perceptual and relational attributes. This pattern suggests that larger models allocate more capacity to abstract conceptual structure, resulting in more coherent projection geometry in domains less tied to physical properties.

## 5   RELATED WORK AND DISCUSSION

**Conceptual Knowledge in Language Models.** Pretrained language models (LMs) have demonstrated remarkable capabilities in capturing conceptual knowledge (Wu et al., 2023; Lin & Ng, 2022). A variety of methods have been developed to probe such knowledge, most commonly binary probing classifiers (Aspillaga et al., 2021; Michael et al., 2020) and hierarchical clustering (Sajjad et al., 2022; Hawasly et al., 2024), with validation against human-defined ontologies such as WordNet (Miller, 1995). These approaches primarily provide empirical insights into what LMs capture, but they leave open the question of how LMs learn such conceptual structures. Moreover, it has been argued that LMs can develop novel concepts not strictly aligned with existing ontologies (Dalvi et al., 2022), suggesting that ontology-based evaluations may underestimate their conceptual capacity. Xiong & Staab (2025) first show the connection of FCA to language models by define concepts in the context of FCA, but their study is limited to masked language models.

**Linear Representation Geometry of Concepts.** A line of research focuses on the geometric nature of concept representations. Several studies suggest that concepts in LMs correspond to distinct directions in activation space (Elhage et al., 2022a; Park et al., 2024a). This linear hypothesis builds on earlier work in distributional semantics and embeddings (Mikolov et al., 2013a; Pennington et al., 2014b; Arora et al., 2016), and has since been connected to multiple notions of linearity, including embedding offsets, probing classifiers, and steering vectors (Park et al., 2024b). Empirical studies have also shown the emergence of polytopes in toy models (Elhage et al., 2022b), pointing to a more structured geometry beyond individual directions. Other theoretical work further explores why linear representations arise, linking them to properties of word embedding models (Arora et al., 2016; 2018) and the implicit bias of gradient descent in LLM training (Jiang et al., 2024).

## 6   CONCLUSION

We presented a new perspective on the geometry of large language models by linking the linear representation of attributes to Formal Concept Analysis (FCA). Our main contribution is the notion of *lattice geometry*, which shows that when attribute directions are treated as separating half-spaces, the resulting object-attribute relation induces a concept lattice. This framework unifies continuous embedding geometry with symbolic abstraction, and provides formal conditions under which logical structure can be recovered from LLM representations. Empirically, we demonstrated on WordNet sub-hierarchies that (i) many attributes are linearly separable, (ii) subsumption can be predicted directly from projection profiles, and (iii) meet and join operations yield meaningful refinements and generalizations. Together, these findings indicate that LLMs encode not only rich conceptual knowledge but also the algebraic backbone of concept lattices. Our results suggest that symbolic abstraction is not an incidental byproduct but a structured geometric property of LLMs. This opens up new directions for neuro-symbolic AI: aligning geometric and symbolic reasoning, enhancing interpretability, and enabling controllable manipulation of concepts in embedding space.

## ETHICS STATEMENT

This work does not involve human subjects or sensitive personal data. Our experiments are conducted on publicly available lexical resources (WordNet) and pretrained LLM embeddings. We follow the ICLR Code of Ethics[1].

## REPRODUCIBILITY STATEMENT

We have made efforts to ensure the reproducibility of our results. The formal definitions, theorems, and proofs are fully detailed in the main text and appendix. Experimental settings, dataset construction (WordNet sub-hierarchies and attribute annotation), attribute direction estimation, and evaluation metrics are described in Section 4. Detailed proofs are provided in the Appendix. Code for reproducing the experiments, including dataset processing and evaluation, are uploaded and will be made available in the supplementary materials.

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

## A    USE OF LARGE LANGUAGE MODELS (LLMS)

Large language models (LLMs) were used as a writing assistant to polish sentences and improve clarity of exposition. Also, we used LLMs to generate annotations of data used for evaluation. No parts of the research ideation, theoretical development, experimental design, or analysis relied on LLMs. All technical contributions, proofs, and empirical results are the work of the authors.

## B    PROOF OF THEOREM 1

We restate the theorem for convenience.

**Theorem 2** (Existence of Lattice Geometry). *Let $G$ be a finite set of objects and $M$ a finite set of attributes. Let $V = \{\mathbf{v}_g \in \mathbb{R}^d \mid g \in G\}$ be the set of object embeddings and $\mathcal{D} = \{\bar{\ell}_m \in \mathbb{R}^d \mid m \in M\}$ the set of attribute directions. Fix $\alpha > 0$. For each $m \in M$, let $\tau_m \in \mathbb{R}$ and define the soft incidence probability*

$$P_\alpha(m(g) = 1) := \sigma\big(\alpha(\mathbf{v}_g \cdot \bar{\ell}_m - \tau_m)\big).$$

*For any confidence level $\delta \in (0, 1)$, define the (crisp) incidence relation*

$$I_\delta := \{(g, m) \in G \times M : P_\alpha(m(g) = 1) \geq \delta\}.$$

*Let $\mathcal{F}_\delta$ be the set of pairs $(X, Y)$ with $X \subseteq G$ and $Y \subseteq M$ such that*

$$X = Y' := \{g \in G : \forall m \in Y, (g, m) \in I_\delta\} \quad and \quad Y = X' := \{m \in M : \forall g \in X, (g, m) \in I_\delta\}.$$

*Then (i) $\mathcal{F}_\delta$ is closed under the Galois connection, and (ii) $\mathcal{F}_\delta$, ordered by extent inclusion (equivalently, reverse intent inclusion), forms a complete lattice.*

**Plan of the proof.**    The probabilistic scoring is only used to induce the crisp relation $I_\delta$. Once $I_\delta$ is fixed, the statement becomes a standard FCA result. For completeness, we give a self-contained proof.

### B.1    GALOIS CONNECTION INDUCED BY $I_\delta$

**Lemma 1** (Antitone Galois connection). *Define maps $(\cdot)' : 2^G \to 2^M$ and $(\cdot)' : 2^M \to 2^G$ by*

$$A' := \{m \in M : \forall g \in A, (g, m) \in I_\delta\}, \qquad B' := \{g \in G : \forall m \in B, (g, m) \in I_\delta\}.$$

*Then for all $A \subseteq G$ and $B \subseteq M$, we have*

$$A \subseteq B' \quad \Longleftrightarrow \quad B \subseteq A'.$$

*Consequently, both primes are antitone: if $A_1 \subseteq A_2$ then $A_2' \subseteq A_1'$, and if $B_1 \subseteq B_2$ then $B_2' \subseteq B_1'$.*

*Proof.* ($\Rightarrow$) If $A \subseteq B'$, then for any $m \in B$ and any $g \in A$ we have $(g, m) \in I_\delta$, hence $m \in A'$ and thus $B \subseteq A'$. ($\Leftarrow$) If $B \subseteq A'$, then for any $g \in A$ and $m \in B$ we have $(g, m) \in I_\delta$, i.e., $g \in B'$, hence $A \subseteq B'$. $\qquad\square$

**Lemma 2** (Closure operators). *The double-prime operators $\phi_G(A) := A''$ on $2^G$ and $\phi_M(B) := B''$ on $2^M$ are closures: for all $A \subseteq G$, (i) $A \subseteq A''$ (extensivity), (ii) $A \subseteq B \Rightarrow A'' \subseteq B''$ (monotonicity), and (iii) $(A'')'' = A''$ (idempotence). Analogous properties hold on $2^M$.*

*Proof.* Extensivity: $A \subseteq A''$ follows from Lemma 1 with $B = A'$: $A \subseteq (A')' = A''$. Monotonicity: $A \subseteq B \Rightarrow B' \subseteq A'$ (antitone), hence $A'' = (A')' \subseteq (B')' = B''$. Idempotence: $(A'')'' = ((A')')'' = (A')' = A''$. The $2^M$ case is symmetric. $\qquad\square$

## B.2 FORMAL CONCEPTS AS CLOSED PAIRS

**Lemma 3** (Characterization of concepts). *A pair $(X, Y)$ with $X \subseteq G$ and $Y \subseteq M$ satisfies $Y = X'$ and $X = Y'$ iff $X$ and $Y$ are closed, i.e., $X = X''$ and $Y = Y''$.*

*Proof.* ($\Rightarrow$) If $Y = X'$, then $X = (X')' = X''$; if $X = Y'$, then $Y = (Y')' = Y''$. ($\Leftarrow$) If $X = X''$, set $Y := X'$ so $X = (X')' = Y'$. The converse from $Y = Y''$ is symmetric. $\qquad\square$

## B.3 LATTICE STRUCTURE AND COMPLETENESS

**Proposition 2** (Partial order). *For formal concepts $(X_1, Y_1)$ and $(X_2, Y_2)$, define*

$$(X_1, Y_1) \leq (X_2, Y_2) \ :\Longleftrightarrow \ X_1 \subseteq X_2 \quad (\text{equivalently } Y_2 \subseteq Y_1).$$

*Then $\leq$ is a partial order on $\mathcal{F}_\delta$.*

*Proof.* Reflexivity/transitivity follow from $\subseteq$. Antisymmetry: $X_1 \subseteq X_2$ and $X_2 \subseteq X_1$ imply $X_1 = X_2$, hence $Y_1 = X_1' = X_2' = Y_2$. $\qquad\square$

**Proposition 3** (Meets and joins). *For any family $\{(X_i, Y_i)\}_{i \in I}$ of formal concepts,*

$$\bigwedge_i (X_i, Y_i) = \left( \bigcap_i X_i, \ \left( \bigcap_i X_i \right)' \right), \qquad \bigvee_i (X_i, Y_i) = \left( \left( \bigcup_i X_i \right)'', \ \bigcap_i Y_i \right),$$

*and both pairs are formal concepts.*

*Proof.* Meet: Let $X_* := \cap_i X_i$. Then $(X_*, X_*') \leq (X_i, Y_i)$ for all $i$. If $(Z, W) \leq (X_i, Y_i)$ for all $i$, then $Z \subseteq X_*$, so $(Z, W) \leq (X_*, X_*')$.

Join: Let $\tilde{X} := (\cup_i X_i)''$ (closed by Lemma 2). Then $(X_i, Y_i) \leq (\tilde{X}, \tilde{X}')$ for all $i$. If $(X_i, Y_i) \leq (Z, W)$ for all $i$, then $\cup_i X_i \subseteq Z$, hence $\tilde{X} \subseteq Z'' = Z$ and $(\tilde{X}, \tilde{X}') \leq (Z, W)$. $\qquad\square$

**Corollary 1** (Completeness). *$(\mathcal{F}_\delta, \leq)$ is a complete lattice: every subset has both meet and join as above.*

## B.4 DISCUSSION OF THE ROLE OF $\alpha$ AND $\delta$

The parameter $\alpha > 0$ only rescales the logits inside $\sigma$ and does not affect order-theoretic conclusions, which depend solely on $I_\delta$. The confidence $\delta \in (0, 1)$ controls the incidence monotonically: if $\delta_1 \leq \delta_2$ then $I_{\delta_2} \subseteq I_{\delta_1}$. Thus increasing $\delta$ removes incidences and yields a different (typically coarser) concept lattice. The theorem holds for each fixed $\delta$.

Table 4: Statistics of the five WordNet-derived datasets.

| Category | WN-Animal | WN-Plant | WN-Food | WN-Event | WN-Cognition |
|---|---|---|---|---|---|
| #Objects | 7342 | 7704 | 2506 | 1009 | 2802 |
| #Attributes | 100 | 145 | 184 | 60 | 107 |
| #Hypernyms | 7473 | 8051 | 2628 | 1079 | 3003 |

## C  PROOF OF PROPOSITION 1

*Proof of Proposition 1.* Let $D \in \mathbb{R}^{k \times d}$ have $i$-th row $\mathbf{d}_i^\top$ and suppose there exists $\mathbf{c} \in \mathbb{R}^d$ with $D\mathbf{c} = \boldsymbol{\tau}$. Then, for each attribute $i \in \{1, \ldots, k\}$,

$$\mathbf{d}_i^\top \mathbf{c} = (D\mathbf{c})_i = \tau_i.$$

Hence, for any object embedding $\mathbf{v}_g$,

$$(\mathbf{v}_g - \mathbf{c}) \cdot \mathbf{d}_i = \mathbf{v}_g \cdot \mathbf{d}_i - \mathbf{c} \cdot \mathbf{d}_i = \mathbf{v}_g \cdot \mathbf{d}_i - \tau_i.$$

Applying the sigmoid with sharpness $\alpha > 0$ yields

$$\sigma\big(\alpha\big(\mathbf{v}_g \cdot \mathbf{d}_i - \tau_i\big)\big) = \sigma\big(\alpha\,(\mathbf{v}_g - \mathbf{c}) \cdot \mathbf{d}_i\big),$$

for all $g$ and $i$, proving that $P_\alpha(m_i(g) = 1)$ is invariant under the global shift $\mathbf{v}_g \mapsto \mathbf{v}_g - \mathbf{c}$. Therefore, the soft-incidence model admits a canonical, origin-passing form without changing any probabilities or the induced incidence relation for a fixed threshold on $P_\alpha$. □

**Remarks.**  (i) The condition $D\mathbf{c} = \boldsymbol{\tau}$ is equivalent to $\boldsymbol{\tau} \in \mathrm{rowspace}(D)$. If the rows $\{\mathbf{d}_i^\top\}_{i=1}^k$ are linearly independent (i.e., $\mathrm{rank}(D) = k$) and $k \leq d$, then $\mathrm{rowspace}(D) = \mathbb{R}^k$, so a (generally non-unique) solution $\mathbf{c}$ exists for any $\boldsymbol{\tau}$. (ii) When solutions exist, they are unique up to addition of any vector in $\ker(D)$; all such choices yield the same invariance because $D(\mathbf{c} + \mathbf{z}) = \boldsymbol{\tau}$ for any $\mathbf{z} \in \ker(D)$.

## D  ADDITIONAL DISCUSSION AND ANALYSIS

**Connection to neuro-symbolic methods and interpretability**    The soft inclusion and meet/join operators introduced in our framework provide differentiable analogues of logical subsumption and concept composition. These scores make it possible to evaluate whether an LLM's embedding geometry behaves logically within a given domain, offering a practical interpretability tool for assessing whether the model has learned coherent concept structure. In addition, because the operators are fully differentiable, they can be used as logic-guided regularizers that softly enforce symbolic constraints during training. Examples include subsumption, attribute satisfaction, and type consistency. This creates a natural interface with neuro-symbolic learning, where symbolic rules influence geometric representations through regularization.

Our findings also suggest a natural extension of linear steering to multi-attribute steering (Oozeer et al., 2025). Whereas standard steering modifies representations along a single attribute direction, lattice geometry enables logical steering. Enforcing two attributes corresponds to moving the embedding toward their meet. Promoting abstraction corresponds to steering toward their join. Negation corresponds to crossing the relevant threshold hyperplane. These operators provide a simple, differentiable, and interpretable way to compose and control conceptual constraints inside LLMs.

Although our experiments focus on WordNet-style taxonomic domains, the framework is broadly applicable to any setting where concepts can be represented by an object–attribute matrix and attribute inclusion induces a partial order. Examples include semantic fields, where words serve as objects and semantic components serve as attributes, and verb classification or speech-act semantics, where verbs are represented by propositional or pragmatic features (Priss, 2005). In a linguistic setting where constructions or grammatical patterns are annotated with syntactic or morphological features, the resulting feature matrix can be treated as an FCA context and modeled through the proposed half-space formulation. In general, any domain that provides attribute-labeled objects admits the lattice geometry developed in this work without requiring modification.

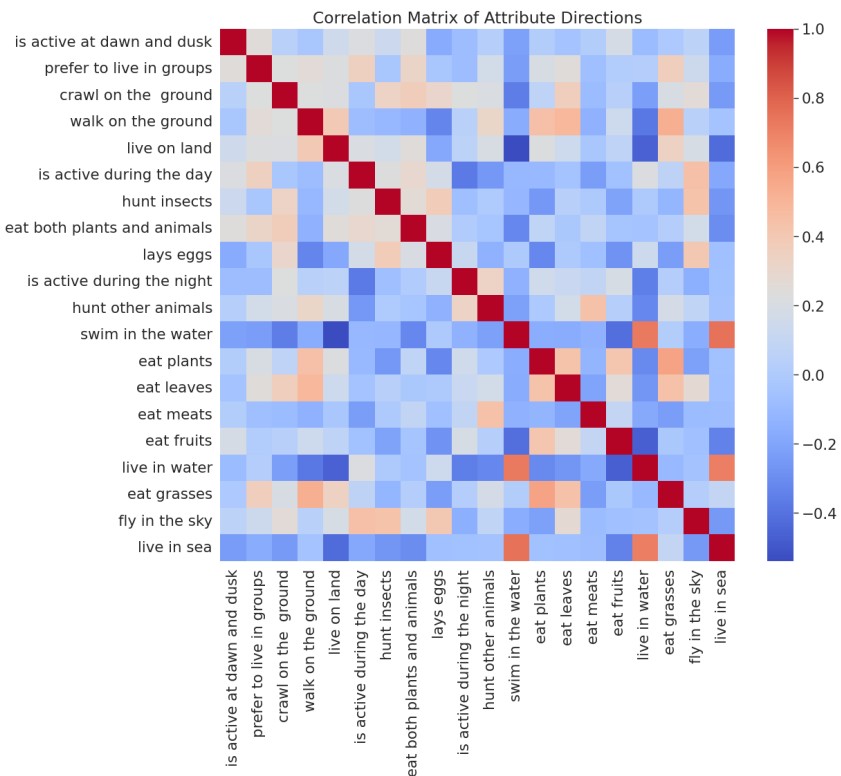

Figure 6: Correlation analysis of attribute directions in the WordNet-Animal dataset.

**Limitation**  Our framework assumes that attributes correspond to approximately linear directions, an assumption supported by prior work on the Linear Representation Hypothesis. However, as noted by Engels et al. (2025), concept types with inherently non-linear, cyclic, or continuously varying structure (for example, months of the year) may violate this linear assumption. Another limitation is that our experimental study focuses on general domains. It is an interesting direction to explore how well the lattice geometry hypothesis holds in domain-specific areas such as biomedical ontologies.

