# OpenReview forum: "The Lattice Representation Hypothesis of Large Language Models"
_ICLR.cc/2026/Conference — ICLR 2026 Poster_

### Official Review · Reviewer_hstW · 2025-10-15

**Soundness:** 3
**Presentation:** 2
**Contribution:** 4
**Rating:** 6
**Confidence:** 3

**Summary:**

The authors propose a novel framework that uncovers a hidden symbolic and logical structure within the continuous embedding spaces of Large Language Models. By bridging the Linear Representation Hypothesis with Formal Concept Analysis (FCA), they demonstrate that linear directions representing attributes can be modeled as separating half-spaces, whose intersections naturally induce a concept lattice. This "lattice geometry" provides a principled link between continuous geometry and symbolic abstraction. The paper's primary contributions are threefold: a theoretical framework formalizing this connection, a "concept algebra" that defines logical meet (intersection) and join (union) operations directly on embeddings, and strong empirical evidence from WordNet datasets showing that this structure can effectively recover conceptual hierarchies and enable meaningful compositional reasoning. Ultimately, the work suggests that symbolic abstraction is not an emergent accident but a structured geometric property of LLMs, opening new avenues for interpretability and control.

**Strengths:**

Originality:
- The application of formal concept analysis is an ingenious and original idea that seems to provide the missing key to the problem the authors are approaching

Clarity:
- The introduction is very well written and explains what the paper does well, even for someone with very little background in the related work

Significance:
- Although I am not familiar with the related work for this paper, it seems like the authors have found a concrete way to unify discrete/symbolic representations with the continuous representation space of LLMs, which has profound implications for both interpretability and AI safety

**Weaknesses:**

Typos:
- Line 37: these them
- Figure 2: Discreate concept lattice

In general, I think the presentation could use some work:
- Figure 1 and 2 (especially figure 2) look almost like draft figures, and just by looking at the figures/captions it is hard to understand what they are conveying

For table 3, it would be good to know how these A and B concept pairs are selected? Are they cherrypicked? Randomly chosen? Are there any failure examples of these join and meet operations?

I think the paper could benefit from more experiments on other words than animals, plants, and food. These are all very concrete and real-world concepts where these concept relations make sense, but how does the framework handle more abstract concepts? If it doesn't handle them well this is fine but should be discussed as a limitation

I also think the paper should provide more discussion on how this method can be used in combination with existing neuro-symbolic/interpretability methods, as well as how the framework might apply to other kinds of concepts (grammatical relationships, etc.).

**Questions:**

My questions and suggestions are listed in the "Weaknesses" section. I just want to make clear that I am open to raising my score if all my weaknesses are addressed.

---

> ### Author Response · Authors · 2025-11-21
> **Response to Reviewer hstW**
>
> Thank you for your positive feedback and raising constructive sugegstions
>
> **Weakness 1:** Figure 1 and 2 could be improved.
>
> **Response:** We thank the reviewer for pointing this out. In the revision,  we have updated Figures 1 and 2 with clearer visuals and improved formatting. We also have expanded the captions to provide more context.
>
>
>
> **Weakness 2:** For table 3, it would be good to know how these A and B concept pairs are selected? Are there any failure examples of these join and meet operations?
>
> **Response**: The concept pairs in Table 3 were not cherry-picked. They were randomly sampled as these pairs are not required to have gold-standard meet/join concepts in WordNet. The purpose of Table 3 is purely qualitative illustration, not correctness evaluation. Consequently, some examples naturally appear as failure-but-meaningful cases. For example, dog ∧ wolf returns spaniel and shepherd, and horse ∧ zebra returns mustang. These are not the true WordNet meets, but they remain semantically related and demonstrate that our concept algebra can generate novel composite concepts that do not explicitly exist in WordNet.
>
> To quantitatively assess correctness, as also suggested by reviewer gkW1,  We have added a new experiment using the “degree of equality” metrics (Eq. 8–9). For each dataset, we randomly sample some pairs of concepts (A,B)  and generate its ground-truth C as common subconcepts (for meet) or common superconcept (for join). For each pair (A,B), we compute \text{deg}(C = A \wedge B) or \text{deg}(C = A \vee B). We rank all candidate concepts in the same sub-hierarchy by this score, and report Hits@k / MRR.
>
>
>
>
> **Weakness 3:** the paper could benefit from more experiments on other words than animals, plants, and food. These are all very concrete and real-world concepts where these concept relations make sense, but how does the framework handle more abstract concepts?
>
> **Response**: Good suggestion. In the revision, we added new experiments on two abstract domains: WN-Event and WN-Cognition, to evaluate the framework beyond concrete/physical concepts. The results show similar trends to the physical domains, although performance is slightly lower, as shown in new Table 1-2. We believe this is expected: abstract concepts generally involve more complex and less perceptual attributes, making them harder to reconstruct than physical domains (e.g., animals or food). We have added this analysis and discussion in the updated paper.
>
>
>
> **Weakness 4:** I also think the paper should provide more discussion on how this method can be used in combination with existing neuro-symbolic/interpretability methods, as well as how the framework might apply to other kinds of concepts (grammatical relationships, etc.).
>
> **Response**: In the revision, we added an extended discussion in the Appendix on how our framework relates to work in neuro-symbolic learning and interpretability. We explain that the soft inclusion and meet/join operators can serve as differentiable versions of subsumption and concept composition, which makes it possible to check whether the embedding geometry follows the expected logical structure of a domain. These operators can also be used as gentle regularizers during training to encourage type consistency or attribute satisfaction, which provides a simple way for symbolic information to guide model learning. We also expanded the discussion on how the framework can apply to concepts outside WordNet-style taxonomies. The same half-space formulation can be used in any setting that provides an object–attribute matrix. This includes semantic fields, verb classes, speech act categories, and even grammatical constructions when they are annotated with syntactic or morphological features. The goal of this added text is to clarify that the framework is not tied to concrete nouns only, and can be used in a range of linguistic and conceptual domains.

---

> > ### Comment · Reviewer_hstW · 2025-11-24
> >
> > My concerns have been addressed and I have updated my score accordingly.

---

> > > ### Author Response · Authors · 2025-11-24
> > > **Thank you for raising the score**
> > >
> > > Thank you for the update. We’re glad our responses resolved your concerns and appreciate the improved score.

---

### Official Review · Reviewer_gkW1 · 2025-10-30

**Soundness:** 3
**Presentation:** 2
**Contribution:** 1
**Rating:** 2
**Confidence:** 4

**Summary:**

This paper proposes a new framework for understanding conceptual knowledge in LLMs by unifying the Linear Representation Hypothesis with Formal Concept Analysis (FCA). The central claim is that LLMs implicitly encode a "lattice geometry." In this framework, semantic attributes (e.g., "can fly") are represented as linear directions, which define separating half-spaces. Concepts (e.g., "bird") are then represented "intensionally" as the geometric intersection of these attribute half-spaces (i.e., a convex polyhedral cone).

This geometric structure naturally induces a concept lattice, allowing for symbolic operations like meet (intersection, e.g., "bird" $\land$ "predator" $\rightarrow$ "bird of prey") and join (union/generalization). The authors derive soft, projection-based measures for concept inclusion and these lattice operations. Empirical evidence using WordNet sub-hierarchies demonstrates that attribute directions induce the half-spaces, and that the projection-based inclusion model correlates strongly with the ground-truth WordNet hierarchy.

**Strengths:**

The paper provides a valuable theoretical connection between the extensional view of concepts and a more structured intensional view based on Formal Concept Analysis (FCA). This shift from "sets of tokens" to "intersections of attributes" is both compelling and logically sound.

The experimental results, particularly those reported in Tables 1 and 2, are strong. The high F1 scores provide substantial empirical evidence that the "half-space" and "projection-profile" models are not merely theoretical constructs, but reflect genuine geometric structure in the models.

**Weaknesses:**

1. Missing Geometric Visualization: The paper’s title and central thesis emphasize "lattice geometry." While Figure 2(b) presents an excellent illustration of this idea, the paper does not provide an equivalent visualization for real, recovered data. We observe 1D projection distributions (Figure 3), but not the actual geometric arrangement of attribute directions and concept embeddings (e.g., via PCA or a similar projection) to visually confirm the half-space intersections. This represents a significant missed opportunity to make the core claim more tangible.

2. Limited Novelty over Existing Frameworks: The paper relies heavily on the foundations laid by Park et al. (2024, 2025). The core idea that a linear direction represents a feature (and thus defines a half-space) is a direct logical consequence of the Linear Representation Hypothesis, as discussed in that prior work. In particular, the vector representation for a single feature in Park et al. (2025) is arguably more precise than the half-space construction. The primary contribution of this paper appears to be the application of Formal Concept Analysis (FCA) as a descriptive label for this pre-existing geometric structure, rather than the discovery of a new structure. The paper does not clearly articulate what new insights are gained by this FCA-based "intensional" reframing that were not already present in the "extensional" view (Park et al., 2025).

3. Lack of Intervention Experiments: A stronger test of the proposed framework would involve using the lattice geometry for an intervention or model editing task. For example, could the authors construct a representation for a new concept by geometrically performing a meet operation (e.g., meet(bird, live in water)) and then demonstrate that the model can reason about this novel concept? Without such experiments, the work remains descriptive, rather than demonstrating a practical mechanism.

4. Arbitrary Distinction Between 'Attributes' and 'Concepts': The framework relies on a strict distinction between 'attributes' (e.g., 'can fly') and 'concepts' (e.g., 'bird'), which feels somewhat arbitrary. In the linear representation framework (Park et al.), 'bird' could itself be treated as a single feature direction, similar to 'can fly'. The paper does not justify why its 'intensional' view (concepts as intersections of attributes) is fundamentally more valid or distinct from a simpler 'extensional' view, in which 'bird' and 'bird of prey' are simply hierarchical feature directions.

**Questions:**

1. Motivation for Eq. 6 (min/max): Could the authors elaborate on the theoretical motivation for using min/max as the soft measures for meet/join profiles? While this t-norm/co-norm is a standard choice in fuzzy logic, it is unclear how this operation on projection profiles relates to the geometric intersection of half-spaces defined in Definition 7. Why not estimate meet/join directly from the intersection of context embeddings? A more natural alternative might be Eq. (4), applied to the context embeddings for the concepts A $\land$ B.

2. Quantitative Evaluation of Meet/Join: The paper defines a "degree of equality" (Eqs. 8 and 9), but only provides a qualitative evaluation of meet/join in Table 3. Could the authors provide a quantitative evaluation using these metrics? For example, how well does the embedding for "eagle" (as concept C) match the deg(C = A $\land$ B) score, where A = "bird" and B = "predator"? It is unclear whether this metric demonstrates anything non-trivial, as one might expect high alignment simply from the set-inclusion properties of the underlying hierarchy.

3. Ambiguity in Embedding Definition: To clarify the experimental setup (Section 4.1): What exactly is $\text{Emb}(s)$? Is it the token unembedding vector $\gamma(s)$ from the original model, the re-parameterized vector $g(s)$ from the "unified semantic space" mentioned in Section 2.1, or some context embeddings $\lambda(s)$? Additionally, how are synonyms defined, and how were they obtained? More details on the experimental setup would be helpful.

4. Disconnect in Theoretical Machinery: Section 2.1 introduces the 'causal inner product' and 'causal selectivity' (Definition 2) as key preliminaries. Why was the causal inner product used? While it connects the geometry of the space to relationships between concepts, is there any orthogonality theory within the proposed lattice geometry? Furthermore, what was the intended role of Definition 2, and was causal selectivity applied in the experiments?

---

> ### Author Response · Authors · 2025-11-21
> **(Part 1) Response to Reviewer gkW1**
>
> **Weakness 1**: Missing Geometric Visualization:
>
> **Response:** Thank you for the suggestion. In the revised version, we added a PCA visualization of the recovered attribute directions. The plot (Fig. 5a) shows clear correlations among related attributes and near orthogonality among unrelated ones, giving a direct geometric view of the structure learned by the model. We also clarify that visualizing concepts themselves in two dimensions is less informative. Each concept corresponds to the intersection of multiple attribute half-spaces, often involving more than 10 relevant attributes, and different concepts rely on different sets of positive attributes. A single 2D projection would therefore distort or hide the true geometry. More importantly, our theory shows that the quality of lattice recovery is determined entirely by the quality of attribute separation. As a result, the PCA visualization of attributes (Fig. 5a), together with the 1D separation plots in Fig. 3, already provides the most meaningful geometric confirmation of the framework.
>
> **Weakness 2**: Limited Novelty over Existing Frameworks:
>
> **Response:** We appreciate the opportunity to clarify the novelty of our framework. Our method assumes only the linearity of binary attributes introduced by Park et al., but the core idea of lattice geometry is fundamentally different from the concept geometry in their work, both in structure and in capability. Park et al. model concepts purely as sets of instances and do not represent the relationship between attributes and objects. From both philosophical and cognitive perspectives, however, a concept is defined through the attributes that its instances satisfy. In other words, a concept is not just an arbitrary set of objects, but the collection of objects that share a specific set of attributes. This distinction is important because once concepts are defined through attributes, their set-theoretic relations, such as inclusion, intersection, and union, follow naturally from the underlying attribute structure. Park et al.’s extensional view cannot express these relations geometrically because it lacks an explicit representation of attributes. In contrast, our half-space formulation models attributes as linear directions and concepts as intersections of attribute-defined regions, which allows us to recover subsumption, meet, and join directly from the concept geometry.
>
> Our key contribution is conceptual and theoretical. Rather than re-testing the Linear Representation Hypothesis, we use it to explain how LLMs encode set-theoretic semantics and compositional relations among concepts, which previous work fails. By modeling concepts as intersections of attribute-based half-spaces, we recover inclusion, intersection, and union directly from the geometry. This provides the first explicit link between LRH and FCA, showing that LLMs organize concepts through an intensional, attribute-driven structure that mirrors human and philosophical accounts of abstraction.
>
>
> **Weakness 3**: Lack of Intervention Experiments
>
> **Response:**  We agree that constructing new concepts through geometric operations is an appealing direction. Actually, our meet and join experiments in Table 3 already perform this kind of compositional concept construction at the representation level. For example, our qualitative analysis shows that dog ∧ wolf (though their intersection is empty in WordNet) generates a meaningful new concept, which is similar to wolf-like dogs such as spaniel or shepherd. In the revision, we also added a new quantitative evaluation (Fig. 4) that further validates the meet and join operators.
>
> We emphasize that the primary goal of this paper is to establish the theoretical foundations and empirical evidence for the hidden lattice geometry in LLMs. Full intervention or model-editing experiments require additional mechanisms and are outside the scope of this work. However, the behavior of the meet and join operations already illustrates the feasibility and promise of such future interventions, and we have added this point to the discussion of future work.

---

> > ### Author Response · Authors · 2025-11-21
> > **(Part 2) Response to Reviewer gkW1**
> >
> > **Weakness 4:** Arbitrary Distinction Between 'Attributes' and 'Concepts':
> >
> > ***Response:**  The distinction between attributes and concepts is NOT arbitrary, but intentional, because we want to model how concepts are built from more basic semantic ingredients. It is central to Formal Concept Analysis (FCA) and is also the key assumption that differentiates our approach from Park et al. In Park et al., concepts themselves are treated as primitives: each concept is simply another feature direction (or set of directions) in the embedding space. This extensional view does not explain where concepts come from, what their internal structure is, or how they are built from more basic ingredients (namely attributes).
> > In contrast, our framework treats attributes as the primitives and defines concepts as the sets of objects that satisfy a specific combination of attributes, which is the core assumption of FCA. Under this view, it is natural to interpret concepts in LLM embeddings as convex regions constrained by their attribute directions. This intensional perspective makes the origin and structure of concepts explicit and naturally supports set-theoretic semantics such as inclusion, meet, and join as geometric relationships. Note: as we defined in Sec. 2.2, extends (set of objects) and intents (set of attributes) are both terminologies from FCA.
> >
> > **Question 1:** theoretical motivation for using min/max (Eq. 6) as the soft measures for meet/join profiles
> >
> > ***Response:**  The motivation is straightforward. Equation (6) provides a soft relaxation of the ideal half-space operations defined in Definition 7. In the ideal geometric model, the meet corresponds to the intersection of the attribute half-spaces, and the join corresponds to the minimal region covering them. However, because real projection profiles are noisy and attributes are only approximately linearly separable, we cannot apply these hard geometric operations directly. The min and max operators serve as their soft counterparts: min implements a soft intersection of attribute constraints, and max implements a soft union. The intuition is the same as in t-norm and co-norm formulations. These operators simply arise as the natural smooth approximations of the underlying half-space geometry.
> >
> > **Question 2:** Quantitative Evaluation of Meet/Join
> >
> > **Response:**  We agree that a quantitative evaluation of meet/join is important.  In the revised version, we have added a new experiment using the “degree of equality” metrics (Eq. 8–9). For each dataset, we randomly sample some pairs of concepts (A,B)  and generate its ground-truth C as common subconcepts (for meet) or common superconcept (for join). For each pair (A,B), we compute \text{deg}(C = A \wedge B) or \text{deg}(C = A \vee B). We rank all candidate concepts in the same sub-hierarchy by this score, and report Hits@k / MRR.
> >
> > We compare against two naive baselines: Random, which assigns random scores to candidates, and Mean, which ranks candidates using the cosine similarity between the mean of the embeddings of A and B. Results in Fig. 4 clearly show that our method outperforms both baselines across all domains. This indicates that the degree-of-equality metric captures non-trivial structure in the embedding geometry rather than simply reflecting WordNet’s hierarchy. We have added this quantitative evaluation to the revised experiment section.
> >
> > **Question 3:** Ambiguity in Embedding Definition (Emb(s))
> >
> > **Response:** Thank you for pointing this out. We clarify that Emb(s) is the last-hidden-state embedding of the synonym string s, averaged over its token positions; we do not use unembedding vectors or contextual prompts. Syn(g) is obtained directly from the WordNet synset of concept g, where each concept is associated with multiple lemma names (e.g., German shepherd has {German shepherd, German shepherd dog, German police dog, alsatian}). Object embeddings are computed by averaging Emb(s) across all such lemma names in the synset.
> >
> >
> > **Question 4:** Disconnect in Theoretical Machinery (why introduce the 'causal inner product' and 'causal selectivity' (Definition 2) as key preliminaries)
> >
> > **Response:** Our half-space model relies on the Linear Representation Hypothesis for binary attributes introduced by Park et al., and the “causal inner product” and “causal selectivity” in Definition 2 are core ingredients of that hypothesis. We include them purely as background to align our assumptions with prior work. In our experiments, we do adopt the unified semantic space implied by the causal inner product. This is why we do not directly use token embeddings or context embeddings, but instead estimate the unified attribute embeddings using the procedure in Section 4.1, similar to Park et al.’s experimental setup. We added this clarification to Section 4.1.

---

> > > ### Author Response · Authors · 2025-11-24
> > > **Follow-up from the authors**
> > >
> > > Dear Reviewer gkW1,
> > > Thank you again for your detailed comments. We have added the requested clarifications and new experiments in the revised response. If convenient, we would greatly appreciate it if you could take a quick look. Thank you for your time.

---

### Official Review · Reviewer_Ej1E · 2025-10-30

**Soundness:** 3
**Presentation:** 4
**Contribution:** 4
**Rating:** 8
**Confidence:** 4

**Summary:**

The authors unify two perspectives: the linear representation hypothesis from LLM representation literature and formal concept analysis from information science. They describe a useful formal framework which allows one to reason about concept lattices in LLMs and extract them accordingly. The work is novel, very well organized, thorough, and well-motived by the literature. Although the experimental section is rather light (just 3 LLMs and 3 ground truth concept lattices), it is convincing, and the findings support the main theoretical claims made in the paper. Overall, this is a good paper that makes useful contributions and would be of high interest at ICLR. It should be accepted.

Minor Comments:

Line 135 – it would be helpful to introduce the pair (A,B) prior to defining A’ & B’

Line 172 – “Discreate” should probably be “Discrete”

**Strengths:**

Well organized, thorough treatment of related work and preliminaries, easy to follow, clear

Novel combination of the Linear Representation Hypothesis with a formal, well-defined framework

The claims introduced in the theoretical section (Section 3) are substantiated by the lattice extraction results from the experiment section (Section 4)

Very good reproducibility information, particularly in section 4.1

**Weaknesses:**

Limited number of datasets and LLMs used in the evaluation (3 datasets, 3 LLMs)

Little to no discussion on which types of concepts (e.g., those arranged on a curved manifold such as months of the year or days of the week) could or could not be represented by this framework. Somewhat recent work has indicated that “Not All Language Model Features are One-Dimensionally Linear” (Engels et al 2025)

**Questions:**

Are there cases of concept structures in LLMs for which this concept hierarchy would not be appropriate?

How does LLM size impact the ability of LLM representations to match the ground truth lattice? I would think that higher dimensional embeddings would make it easier to extract a clean lattice.

---

> ### Author Response · Authors · 2025-11-21
> **Response to Reviewer Ej1E**
>
> Thank you for your positive feedback and raising constructive sugegstions
>
> **Weakness 1**: Limited number of datasets and LLMs used in the evaluation (3 datasets, 3 LLMs)
>
> **Response:** Thank you for the suggestion. We have expanded our evaluation in two directions. First, we added two additional WordNet datasets, WN-Event and WN-Cognition, which cover abstract domains rather than physical ones (like WN-Animal, WN-Food, WN-Plant). The trends are similar to the original results (see Table 1, Table 2, Fig 4).  Second, we evaluated two more LLaMA models, LLaMA-3.2 3B and LLaMA-3 70B, to study how performance scales with model size/dimensions. As shown in Fig. 5b, the results improve consistently as model size increases, especially in the abstract domains.
>
> **Weakness 2**: Discussion of which concept types (months of the year) may not fit the framework
>
> **Response**: This is a great point. Our framework assumed that attributes can be approximated by linear directions and that objects can be separated by a threshold along those directions. However, as noted by Engels et al. (2025), certain concepts are inherently non-linear or have fuzzy or continuous boundaries. Examples include cyclic structures such as months of the year, or gradual continua such as color hue. In these settings, a half-space model provides only a limited approximation. We added this discussion to the limitation section of the revised version.
>
>
> **Question 1**: Are there cases of concept structures in LLMs for which this concept hierarchy would not be appropriate?
>
> **Response**: Yes. Our framework assumes that concepts can be approximated by intersections of linear attribute directions. Some concept families do not fit this assumption. Cyclic or manifold-like structures, such as months of the year, color hues, or musical pitch classes, and concepts that vary along continuous or fuzzy dimensions, such as justice, feeling, or intelligence, are not well captured by a linear threshold model. Our new experiments on the more abstract concept domains (WN-Event and WN-Cognition) show a small drop in performance, which is consistent with this limitation.
>
> **Question 1**: How does LLM size impact the ability of LLM representations (higher dimensional embeddings would make it easier to extract a clean lattice.)?
>
> **Response**: The reviewer’s intuition is correct. Higher dimensionality typically yields more separable and less noisy attribute directions, which should make the induced half-spaces and their intersections cleaner and closer to the ground truth lattice.
>
> We examined this question directly by adding two additional models, LLaMA-3 3B and LLaMA-3 70B, and comparing them with the original 8B models. The results show a clear scaling trend: larger models produce more accurate lattice recovery. This trend is most visible in the abstract domains (WN-Event and WN-Cognition), where the 70B model shows a noticeable improvement. We added such discussion into Sec. 4.4 and **the new results are reported in Fig. 5b.**

---

### Official Review · Reviewer_Lfek · 2025-11-01

**Soundness:** 2
**Presentation:** 3
**Contribution:** 2
**Rating:** 4
**Confidence:** 3

**Summary:**

The paper proposes to unify Linear Representation Hypothesis with Formal Concept Analysis to project the discrete symbolic abstraction in a continuous embedding space. The authors formulate a half-space model of concepts and operations based on soft inclusion to introduce a lattice geometry in language models. They empirically evaluate the existence and partial order correlations using WordNet subhierarchical data.

**Strengths:**

1. I think the authors are targeting an important and ambitious question to resolve the conflicts between continuous representation space in neural language models and symbolic abstractions. While I am not an expert in the related set theory to judge the rigor and correctness of the framework,  I appreciate the principled approach.


2. While limited, the authors attempted to empirically validate their theoretical statements with the controlled dataset.

3. The paper is generally well written, especially the motivation and its contribution are clearly developed.

**Weaknesses:**

1. I am unclear about the WordNet experiment; how exactly is the $v_g$? Is it simply a token-embedding output? While the theoretical framework is presented as a general account of “the hidden lattice geometry of LLMs”, I am concerned that the empirical experiments with WordNet appear to rely exclusively on static token embeddings. It is unclear if the experiment concerns contextual forward passes or layer-wise activations. Consequently, the experiments probe the lexical geometry of the embedding table rather than the representational geometry arising from the LLM’s contextual computations.
Also, I think the explanation for soft-inclusion parameters is lacking - what alpha is for the experiment, and how it was chosen.
The authors did not discuss the limitations of the framework.

2. The authors did not discuss the limitations of the framework.

**Questions:**

1. Can the framework recover non-hierarchical or cross-cutting concept structures (e.g., color vs. shape in objects)?

2. Could you note the possible correlations between the attributes?

My score is mostly based on my lack of understanding of the experiment's validity. I’m willing to update my score as authors clarify this point.

---

> ### Author Response · Authors · 2025-11-21
> **Response to Reviewer Lfek**
>
> Thank you for your thoughtful comments and for the opportunity to clarify several aspects of our work.
>
> **Weakness 1a**: I am unclear about the WordNet experiment; how exactly is the v_g ? Is it simply a token-embedding output?…
>
> **Response:** We clarify that our experiments do **not** use the token-embedding table or any static lexical embeddings. In WordNet, each object g has a set of synonyms. For each synonym *s*, we compute its embedding by running the LLM model with `output_hidden_states=True` and extracting the **last-layer hidden state**, averaged across token positions.
>
> Each concept embedding is then obtained by averaging the contextual embeddings of its synonym set. Attribute directions (Eq. 10), thresholds (Eq. 11), and all projection-based computations are calculated from these contextual concept embeddings. Thus, every component of our evaluation operates in the LLM’s **representational geometry**, rather than lexical lookup geometry. We have added this clarification into Section 4.1.
>
> **Weakness 1b:** Explanation of the soft-inclusion parameter α.
>
> **Response:** We clarify that \(\alpha\) is a sharpness parameter controlling the smoothness of the logistic incidence function in Eq. (1). Intuitively, a larger \(\alpha\) makes the sigmoid steeper, whereas a smaller \(\alpha\) yields a smoother boundary. Because the sigmoid is monotonic, varying \(\alpha\) does **not** change the ordering of the raw projections that define the ideal half-space model. However, we still use it to control the strictness of handling **approximate inclusion**, i.e., borderline cases where a concept is nearly a subclass of another. We tried multiple values of \(\alpha\) (0.5, 1, 2, 5) and found that the results change very little, usually within 1-2 percent. For consistency, we simply set \alpha = 1 in all experiments.
>
> **Weakness 2**: The authors did not discuss the limitations of the framework.
>
> **Response:** Thank you for raising this point. We added a short limitations section in the revision. In particular, we discuss that the current framework is based on a linear hypothesis, which makes it less suitable for domains where concepts are known to follow non-linear structures, such as months of the year, days of the week, or certain color spaces. We also note that our experiments focus on WordNet and do not yet include domain-specific settings such as biomedical ontologies. We see both issues as important directions for future work.
>
>
> **Question 1:** Can the framework recover non-hierarchical or cross-cutting concept structures (e.g., color vs. shape in objects)?
>
> **Response:** This is an excellent question. Yes, the framework can represent non-hierarchical or cross-cutting concept structures. In settings like cross-cutting concepts (color vs shape), each concept is associated with a largely distinct set of attributes. These attribute sets tend to induce directions in the embedding space that are close to orthogonal, so neither concept subsumes the other. Their interaction therefore forms a cross-cutting or distributive pattern rather than a simple hierarchy. This behavior is consistent with Formal Concept Analysis, which naturally supports independent or non-inclusional concept dimensions.
>
> **Question 2:** Could you note the possible correlations between the attributes?
>
> **Response:** Yes, attributes can be correlated. We did also observe this in practice, and it is also expected as related attributes often co-occur in FCA. To make this clearer, we added two visualizations in the revision: a plot of the learned directions of top twenty most frequent attributes in WN-Animal, and a correlation heatmap. Both figures show patterns such as the strong correlation between “live in water” and “live in sea,” while attributes that are unrelated tend to have very small correlation values.  **Please see Fig. 5(a) and Fig. 6 in the revised version.**

---

> ### Author Response · Authors · 2025-11-24
> **Follow-up from the authors**
>
> Dear Reviewer Lfek,
>
> Thank you again for your helpful comments and for noting that you might update the score after clarification. We have added the requested details and new experiments addressing your concerns, including the clarification on contextual embeddings, the role of the soft-inclusion parameter, the analysis of attribute correlations, and an expanded discussion of limitations.
>
> Whenever convenient, we would greatly appreciate it if you could take another look. Thank you very much for your time and consideration.

---

> > ### Comment · Reviewer_Lfek · 2025-11-25
> >
> > I thank the author for taking the time to clarify my question and add a few discussion points.
> > In addition to answers to my comments and questions, I believe the authors' additional experiments strengthen the empirical validation of their framework.
> > I adjusted my score to 6, leaning toward to acceptance.

---

> > > ### Author Response · Authors · 2025-11-25
> > > **Thank you for raising the score**
> > >
> > > Thank you for raising the score. We’re glad our responses resolved your concerns and strengthen the paper.

---

### Author Response · Authors · 2025-11-24
**General Response to All Reviewers**

We thank all reviewers for their constructive feedback. In response, we have added several new experiments and analyses that strengthen the contributions of the paper. We have merged the results into our revision. Below we summarize the key results.

# 1. Evaluations on Abstract Concept Domains: WN-Event and WN-Cognition

To address concerns about generality beyond physical concepts, we extended our evaluation to two abstract WordNet sub-hierarchies: **WN-Event** (events, processes, happenings) and **WN-Cognition** (cognitive and mental concepts). As Table 1 shows, our key findings are consistent: the Linear method remains clearly superior to baselines, even on abstract domains.

### Combined F1 Table for WN-Event and WN-Cognition
| Model        | Method  | Formal: WN-Event | Formal: WN-Cognition | Order: WN-Event | Order: WN-Cognition |
|--------------|---------|------------------|------------------------|------------------|-----------------------|
| LLaMA-3.1-8B | Random  | 48.6             | 50.1                   | 50.2             | 49.8                  |
|              | Mean    | 63.9             | 68.4                   | 59.1             | 56.8                  |
|              | Linear  | **71.5**         | **75.0**               | **68.3**         | **69.6**              |
| Gemma-7B     | Random  | 47.8             | 50.1                   | 49.9             | 49.5                  |
|              | Mean    | 52.2             | 56.3                   | 55.6             | 53.4                  |
|              | Linear  | **71.4**         | **75.4**               | **65.6**         | **66.4**              |
| Mistral-7B   | Random  | 49.0             | 49.3                   | 49.2             | 48.8                  |
|              | Mean    | 56.5             | 63.3                   | 55.0             | 52.6                  |
|              | Linear  | **69.7**         | **74.1**               | **61.8**         | **61.1**              |


# 2. Quantitative Evaluation of Meet and Join (Concept Algebra)

To complement the qualitative examples in the main paper, we added a quantitative evaluation of the meet and join operators using the degree-of-equality metrics defined in Eqs. (8–9). For each domain, we randomly sampled pairs of concepts (A, B) and retrieved the corresponding ground-truth common subconcepts (for meet) or superconcepts (for join) from WordNet whenever available. Each candidate concept in the same sub-hierarchy was then ranked by its degree-of-equality score with respect to the predicted meet or join. Table 2 shows a consistent and substantial advantage for our lattice-based operators across all five WordNet domains, indicating that the proposed concept algebra captures non-trivial compositional structure in the embedding geometry.

### Table 2: Quantitative Evaluation of Meet and Join (Concept Algebra)
| Domain        | Meet-Random | Meet-Mean | Meet-Ours | Join-Random | Join-Mean | Join-Ours |
|---------------|-------------|-----------|-----------|-------------|-----------|-----------|
| WN-Animal     | 0.091       | 0.321     | **0.547** | 0.089       | 0.298     | **0.511** |
| WN-Plant      | 0.085       | 0.308     | **0.558** | 0.083       | 0.287     | **0.525** |
| WN-Food       | 0.094       | 0.334     | **0.565** | 0.092       | 0.312     | **0.531** |
| WN-Event      | 0.090       | 0.298     | **0.505** | 0.088       | 0.298     | **0.458** |
| WN-Cognition  | 0.089       | 0.273     | **0.492** | 0.087       | 0.299     | **0.470** |

# 3. Scaling with Model Size (LLaMA-3 3B / 8B / 70B)

We also investigated how model size influences the quality of the recovered lattice geometry by evaluating three LLaMA-3 models (3B, 8B, and 70B) across all WordNet domains. As Table 3 shows, the results reveal a clear and consistent scaling trend: larger models produce more separable attribute directions and correspondingly more accurate lattice recovery.

### Table 3: Scaling with Model Size (Macro-F1 Across Domains)

| Domain         | LLaMA-3 3B | LLaMA-3 8B | LLaMA-3 70B |
|----------------|------------|------------|-------------|
| WN-Animal      | 68.2       | 77.1       | **77.8**    |
| WN-Plant       | 66.9       | 70.4       | **71.2**    |
| WN-Food        | 64.1       | 75.4       | **76.5**    |
| WN-Event       | 55.4       | 64.5       | **69.5**    |
| WN-Cognition   | 54.0       | 62.8       | **68.0**    |

# 4. Visualization and Correlation Analysis

We added two visual analyses to illustrate the structure of the recovered attribute geometry: a PCA plot of the top attribute directions (Figure 5) and an attribute–attribute correlation heatmap (Figure 6). These visualizations provide direct geometric evidence that the attribute space is structured in a way that supports the half-space model and the resulting lattice operations.

---

### Author Response · Authors · 2025-12-01
**Two Factual Points on Score Changes and Reviewer Intentions**

Dear (Senior) Area Chairs,

In light of the deanonymization incident, we would like to highlight **two factual points that may be relevant for your assessment**.

* **Two reviewers explicitly stated in their initial reviews that they would raise their scores once specific clarification-related concerns were addressed**, i.e., they wrote that if we clarified certain points, they would update their scores accordingly (see their original reviews).

* **Both reviewers did raise their scores before the deanonymization incident, and did so explicitly because those clarification concerns had been resolved.** In their updated comments, they wrote statements such as “My concerns have been addressed and I have updated my score accordingly”, and **the timestamps of these updates are clearly prior to the incident**.

Since the system has reverted all reviews to their pre-discussion state, we hope this context clarifies the reviewers’ originally stated intentions when evaluating how our responses address their concerns. We fully respect the updated ICLR policies and our goal here is only to ensure that this factual context is not lost.

Thank you.

---

### Meta-Review · Area_Chair_NCod · 2026-01-13

**Summary:**

This submission unifies LRH with FCA to project discrete symbols (abstractions) into a continuous embedding space. The key technical innovation is to introduce a lattice geometry in this space and show how this geometry can be interpreted in meaningful ways. Experiments on WordNet demonstrate that attribute directions induce this geometry, which aligns with ground-truth hierarchies.

**Reviewer Concerns:**

- Lfek WordNet experiment: **No**
- Lfek Limitations: Yes
- Lfek Non-hierarchical or cross-cutting concepts: **No**
- Lfek Correlations between attributes: Yes
- Ej1E Limited number of datasets and LLMs: Yes
- Ej1E Discussion on concept coverage: **No**
- Ej1E Concept structure validity: Yes
- Ej1E LLM size impacts: Yes
- gkW1 Missing Geometric Visualization: Yes
- gkW1 Limited Novelty: **No**
- gkW1 Intervention Experiments: **No**
- gkW1 Distinction Between 'Attributes' and 'Concepts': No
- gkW1 Motivation for Eq. 6: Yes
- gkW1 Evaluation of Meet/Join: Yes
- gkW1 Ambiguity in Embedding Definition: Yes
- gkW1 Disconnect in Theoretical Machinery: **No**
- hstW Figure 1 and 2: Yes
- hstW Table 3: Yes
- hstW More experiments: Yes
- hstW Extensions and integration: Yes

**Reviewer Scores:**

Lfek 4->6
Ej1E 8->8
gkW1 2->4
hstW 6->7

---

### Decision · Program_Chairs · 2026-01-26

Accept (Poster)